# ConnectomeBench: Can LLMs Proofread the Connectome?

**Jeff Brown**
MIT
jffbrwn@mit.edu

**Andrew Kirjner**
MIT

**Annika Vivekananthan**
MIT

**Ed Boyden**
HHMI
Yang Tan Collective
McGovern Institute
MIT Departments of Brain and Cognitive Sciences

## Abstract

Connectomics—the mapping of neural connections in an organism's brain—currently requires extraordinary human effort to proofread the data collected from imaging and machine-learning assisted segmentation. With the growing excitement around using AI agents to automate important scientific tasks, we explore whether current AI systems can perform multiple tasks necessary for data proofreading. We introduce ConnectomeBench, a multimodal benchmark evaluating large language model (LLM) capabilities in three critical proofreading tasks: segment type identification, split error correction, and merge error detection. Using expert annotated data from two large open-source datasets—a cubic millimeter of mouse visual cortex and the complete Drosophila brain—we evaluate proprietary multimodal LLMs including Claude 3.7/4 Sonnet, o4-mini, GPT-4.1, GPT-4o, as well as open source models like InternVL-3 and NVLM. Our results demonstrate that current models achieve surprisingly high performance in segment identification (52-82% balanced accuracy vs. 20-25% chance) and binary/multiple choice split error correction (75-85% accuracy vs. 50% chance) while generally struggling on merge error identification tasks. Overall, while the best models still lag behind expert performance, they demonstrate promising capabilities that could eventually enable them to augment and potentially replace human proofreading in connectomics. Project page and Dataset

.

## 1  Introduction

Recent advances in large language models (LLMs) have sparked interest in their application to complex scientific tasks. While these models demonstrate increasingly sophisticated reasoning capabilities in math and software, their multimodal visual reasoning abilities have also shown particularly impressive gains. For example, OpenAI's o3 model now approaches human-level performance on visual reasoning over scientific charts (see CharXiv, Wang et al. [2024]). The potential for human-level visual reasoning capabilities represents an opportunity to address bottlenecks in time intensive tasks in science that rely heavily on human perception and judgment.

Connectomics—the comprehensive mapping of neural connections in an organism's brain—represents a compelling test case for such capabilities. Creating a connectome begins with high resolution

imaging of brain tissue to create an image volume, followed by computational segmentation to identify individual components within the volume like neurons and their processes. Unfortunately, even state-of-the-art segmentation algorithms produce systematic errors that require human correction. As such, the manual "proofreading" process to correct these errors represents a significant bottleneck in connectome creation. For example, Dorkenwald et al. [2024] report that the first complete reconstruction of a fruit fly connectome required an estimated 33 human years of manual proofreading effort. If efforts to scale to larger brain connectomes are going to be feasible, new methods for automated connectome proofreading are essential. One potential avenue could be through AI agent systems capable of proofreading data at expert-level quality.

To explore if LLMs can provide a path toward automated proofreading, this paper introduces ConnectomeBench, a multimodal benchmark designed to evaluate the performance of LLMs on three fundamental proofreading tasks:

1. Segment type identification: Classifying segmented structures as single neurons, merged neurons, neuronal processes without soma, nuclei, or non-neuronal cells.

2. Split error correction: Determining whether two separated segments should be merged as part of the same neuron.

3. Merge error identification: Detecting when segments from multiple neurons have been incorrectly combined.

For each task, we develop evaluations grounded in data from two major open-source connectome datasets: a cubic millimeter of mouse visual cortex The MICrONS Consortium [2025] and the complete drosophila brain Dorkenwald et al. [2024]. Our benchmark leverages the multimodal capabilities of LLMs, presenting them with images of 3D segmentation data and assessing their performance through both binary classification and multiple choice evaluations.

ConnectomeBench offers several contributions to the scientific community. First, it provides a standardized method for evaluating LLM capabilities in connectome proofreading, allowing for consistent comparison across models and over time. Second, it establishes a performance baseline for current frontier models on these tasks, identifying both current capabilities and limitations. Finally, it creates a foundation for developing specialized LLM-based agents that could one day remove the human effort currently required for connectome creation.

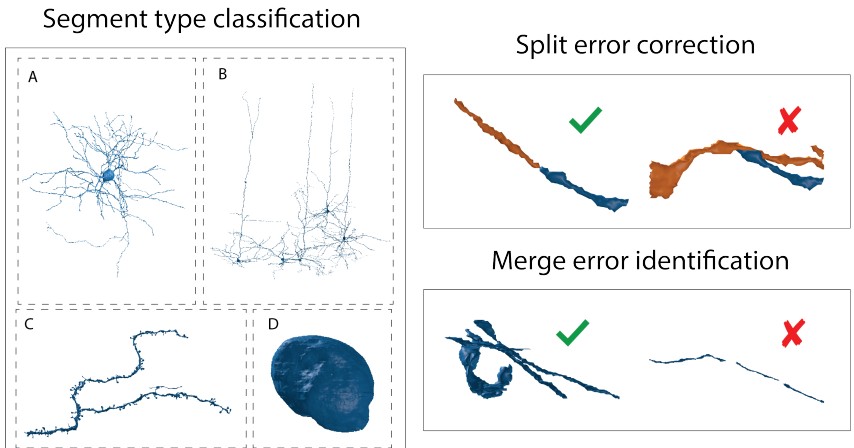

Figure 1: Summary of the three tasks evaluated in ConnectomeBench. In the left panel are examples of four different types of segments: A) single neuron, B) multiple neurons merged together, C) neuronal processes with a cell body (soma), and D) isolated cell nucleus. Examples of 3D segments of non-neuronal cell types can be found in Appendix A. In the right upper panel is an example of segment with a split error (in blue) and two potential merge candidate to correct the error (in orange). On the left is a correct merge candidate; on the right is an incorrect merge candidate. In the right bottom panel are examples of segments with and without merge errors (on the right and left respectively).

# 2 Prior work

The field of automated connectome proofreading has evolved significantly over the past decade, with researchers developing diverse computational approaches to address the bottleneck of manual error correction.

Several methods leverage heuristics over graph representations of 3D segmentation data to identify and correct segmentation errors. Joyce et al. [2023] employed mesh processing techniques to identify jagged areas near neuronal tips that likely indicate false splits. Celii et al. [2025] created NEURD, which transforms 3D neuron meshes into annotated graph representations and uses heuristic graph rules for automated merge error correction. While heuristics can work well and are richly interpretable, they can also be quite brittle, which can be challenging in the irregular data environment of proofreading. To meet the needs of flexibility, deep learning has been used to significantly advance the field of connectome proofreading. Early work by Haehn et al. [2017] introduced guided proofreading using convolutional neural networks (CNNs) to recommend candidate merge and split operations to users. Li et al. [2020] developed a method to classify neuron subcompartments (axon, dendrite, soma) using 3D CNN models, leveraging these predictions to detect and correct merge errors. Schmidt et al. [2024] developed RoboEM, a CNN that traces processes (e.g. axons, dendrites, neurites) throughout the volume by treating it as flight-steering problem. Troidl et al. [2025] introduced point affinity transformers to process point clouds derived from the 3D segmentation to automatically identify merge errors through clustering.

With this context, we sought to understand if multimodal LLMs could leverage the best of both worlds — heuristic interpretability and processing flexibility — along with their prior knowledge to perform proofreading tasks. If so, this could pave the way for AI agent based connectomics proofreading. Nguyen et al. [2021] established a precedent for agent-based connectome proofreading with RLCorrector. RLCorrector used reinforcement learning agents for detecting, classifying, and correcting both merge and split errors. This approach demonstrated how an agent system could model the human proofreading workflow, making decisions at each step based on learned policies rather than fixed rules or supervised models alone. While early, this work anticipates the opportunity to build AI agents for connectomics proofreading.

# 3 Dataset Construction

## 3.1 Background

In creating a connectome, first an organism's brain is imaged using a high-resolution imaging technique like electron microscopy (EM) or, more recently, expansion microscopy Tavakoli et al. [2025] to produce many "slices." Each slice contains an image of stained brain tissue at nanometer resolution. These slices are computationally aligned and stacked together to produce an imaging volume. Afterward, a segmentation algorithm is applied to the volume data to generate three-dimensional segmentations intended to isolate individual components like neurons, non-neuronal cells, and blood vessels. In practice, both the data and segmentation algorithms are imperfect and this leads to errors in the segmentations. On the data side, there are occasionally issues introduced during imaging or handling where imaging slices are missing, marred, or containing ambiguous signals (due to variable staining or distortion). These issues, coupled with mistakes introduced by the segmentation algorithms, lead to *split errors*, where segments of a single neuron are incorrectly separated, and *merge errors*, where segments from multiple neurons are inappropriately combined. As such, after the initial round of segmentation, human scientists "proofread" the connectome, checking for and correcting segmentations errors.

During manual proofreading, expert annotators examine the imaging and segmentation data in a graphical user interface (GUI) specifically designed for proofreading connectomics data (see Google Inc. [2016]). These GUIs enable one to visualize both the imaging and segmentation data, select and deselect multiple segments, translate and rotate segments in three dimensions, and manually introduce edits to resolve merge or split errors. Due to the contributions of major proofreading campaigns, there are two large open-source connectomics datasets that have undergone the full pipeline of imaging, alignment, segmentation and human proofreading.

- MICrONS: A cubic millimeter of mouse visual cortex by the MICrONs Consortium containing ~200,000 proofread neurons [The MICrONS Consortium, 2025].
- FlyWire: The complete Drosophila brain with ~140,000 proofread neurons [Dorkenwald et al., 2024].

## 3.2 Data Generation

In attempting to gauge LLMs' capability in proofreading, we need both the ability to generate ground truth data about proofreading actions and provide LLMs access to the data necessary for proofreading. Fortunately, both proofread datasets are accessible via the CAVEClient, a Python interface introduced by Dorkenwald et al. [2025] that stores the edit history of each segmentation. Using CAVEClient, one can access the status of the segmentation at every moment of manual proofreading. This includes the initial segmentation results without any proofreading, and every human generated edit that contributed to the final proofread connectomes. We use this client to generate ground truth data to evaluate proofreading capabilities.

We opt to provide LLMs access to the data by prompting them with images of the 3D meshes. To do so, we wrote software to load and save images of the 3D meshes and to provide these images to LLMs through prompts during various tasks. By directly working on the images, the LLMs are able to interact with the data in a way similar to how humans interact with the proofreading GUIs. For every 3D mesh we generate, we generate three viewing angles corresponding to the top, side, and front view of the meshes. Depending on the tasks, we apply a bounding box to crop the 3D mesh. To keep the resolution consistent, every image generated is constrained to 1024 by 1024 pixels. Importantly, we did not train any of the models to recognize or process the images of the 3D meshes; instead, we sought to characterize the models' baseline understanding of the images.

## 3.3 Use of proprietary and open source multimodal LLMs

In this work, we use OpenAI's o4-mini, GPT-4.1, and GPT-4o and Anthropic's Claude Sonnet 3.7 and 4, accessed via their respective APIs, and leave the sampling settings to their default. For the OpenAI models, we set the image detail setting to "high." To account for variability in model response, unless otherwise specified, we run each prompt multiple times (between 5-10) and choose the most common answer for analysis. In addition to providing an answer, the LLMs are prompted to provide their reasoning for downstream analysis.

Alongside these proprietary models, we evaluated open-weight alternatives on a subset of the tasks. Our primary focus for open-weight models was NVIDIA's NVLM and the InternVL-3 family as a representative of leading open-weight multimodal architectures. Our InternVL experiments included two versions: the InternVL-3 8B (8 billion parameters) and the significantly larger InternVL-3 78B (78 billion parameters). All computations for these models were performed on a system equipped with 4 NVIDIA H100 GPUs. The computational time for each of the open-source model evaluations was approximately 2-4 hours.

# 4 Tasks and Evaluations

## 4.1 Segment Identification

The first task that we evaluate is whether LLMs can recognize and describe segment types from their 3D meshes. For this task, we group the segments into the five categories in Table 1

Table 1: Distribution of categories for FlyWire and MICrONS datasets

|  | FlyWire | MICrONS |
| --- | --- | --- |
| Single soma and processes | 117 | 130 |
| Multiple somas (and processes) | 13 | 27 |
| Neuronal processes without a soma | 175 | 116 |
| Nucleus | 27 | 92 |
| Non-neuronal types (e.g. glial cells, blood vessels) | 18 | 1 |

You are an expert at analyzing neuronal morphology. In the images, we have a selected 3D segmentation that is supposed to correspond to a complete neuronal structure. However, it could have split/merge errors as the segmentation algorithm makes mistakes.

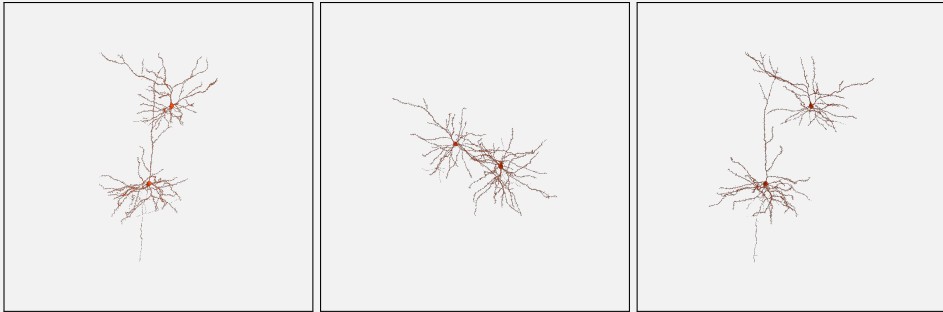

The 3D snapshots are three different views of the same segment. Describe in detail what you see using the information in the 3D snapshots. Is the segment a neuron (soma and processes)? Multiple neurons merged together (multiple somas)? Processes like axon and dendrites without a cell body? Non-neuronal structures like glia, astrocytes, or blood vessels? For mouse neurons, the somas tend to be round and generally a multiple processes extend from them outwards. Processes can be axons or dendrite, long and often branching. Synapses can also be considered as a part of processes, and these are often small segments (smaller than a cubic micron). The nucleuses are round and do not have any processes extending from them. Blood vessels are tubular and obviously do not have any processes extending from them. Glial cells lack the branching processes of neurons, and instead appear like jagged masses.

Choose the best answer:
a) A single soma and process(es).
b) Multiple somas (and processes)
c) Processes without a soma. These can be axons, dendrites, synapses.
d) Nucleus.
e) Non-neuronal types. These can be glial cells, blood vessels.
f) None of the above.

Figure 2: Example prompt used for classifying the segment type. Text in blue is the additional context provided in the "Description" prompts. In this case, the correct answer would be (b).

To get labels for each segment, we generated images of the complete 3D mesh from three perspectives (see example images in Figure 2). Afterward, trained undergraduates and graduate students went through each example to classify and describe the 3D meshes. Then, we evaluate how well the LLMs agree with human judgments. When prompting the proprietary LLMs, we explore two prompting strategies: "Description" where we provide a few sentences describing what different categories look like, and "Null" where we give no additional context (Figure 2). For the open source models, we use the "Description" prompt.

In Table 2, we provide the balanced accuracy (i.e. the average recall across classes) results for each model and dataset. Since we only have one instance of non-neuronal type from the examples pulled from MICrONS, we drop this category from its balanced accuracy calculation. As a baseline, we also include the accuracy of a fine-tuned ResNet classifier (see details in Appendix G). Each proprietary model performs far above the null balanced accuracy (0.2 or 0.25 from randomly choosing one of the available categories). Claude 3.7 Sonnet is by far the best performer in classifying meshes in the MICrONS dataset, while GPT-4.1 and o4-mini are similarly high performers on the FlyWire dataset. Interestingly, providing additional context by describing the categories of 3D meshes does not often improve performance of the proprietary models, suggesting that these models' internal priors already capture the information necessary to identify neuronal components. While decidedly worse than the

Table 2: Balanced accuracy for segment identification task

| Model | FlyWire (95% CI) | MICrONS (95% CI) |
|---|---|---|
| Claude Sonnet 3.7+Description | 0.459 (0.440, 0.480) | **0.822 (0.800, 0.847)** |
| o4-mini+Description | 0.511 (0.477, 0.547) | 0.728 (0.708, 0.747) |
| GPT-4.1+Description | **0.529 (0.495, 0.563)** | 0.655 (0.631, 0.679) |
| GPT-4o+Description | 0.396 (0.373, 0.419) | 0.588 (0.568, 0.610) |
| InternVL-3+Description | 0.320 (0.230, 0.402) | 0.493 (0.440, 0.549) |
| InternVL-3-8B+Description | 0.303 (0.244, 0.376) | 0.417 (0.370, 0.461) |
| NVLM+Description | 0.234 (0.219, 0.250) | 0.258 (0.243, 0.274) |
| Claude Sonnet 3.7 | 0.439 (0.422, 0.455) | 0.819 (0.795, 0.843) |
| o4-mini | 0.476 (0.444, 0.508) | 0.727 (0.707, 0.747) |
| GPT-4.1 | 0.438 (0.412, 0.463) | 0.631 (0.609, 0.654) |
| GPT-4o | 0.337 (0.317, 0.359) | 0.551 (0.533, 0.572) |
| ResNet-50 | 0.552 | 0.587 |

proprietary models, InternVL-3-8B and InternVL-3 perform above the null baseline for both datasets. The per-category accuracy, precision, and recall across conditions is available in Appendix D.

## 4.2 Split Error Correction

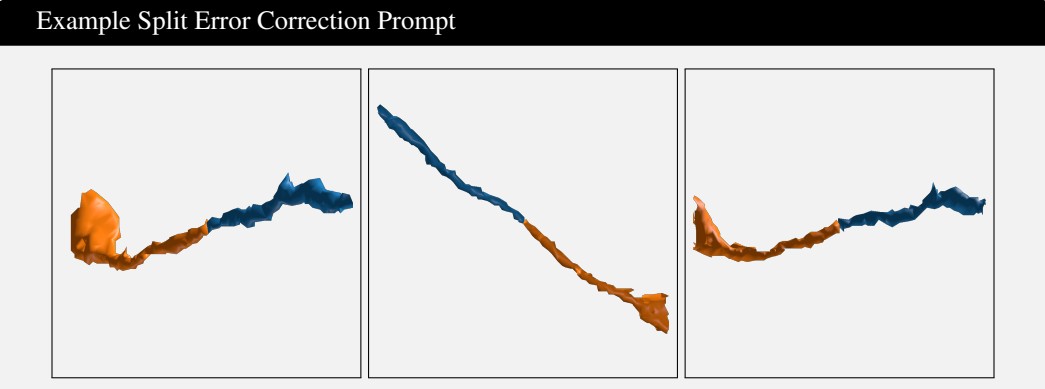

**Example Split Error Correction Prompt**

You are an expert in analyzing neuronal morphology and split and merge errors in connectomics data. The previous images show a proposed merge operation at the center of the 3D volume. The original segment is blue and a potential merge candidate segment is orange. The images below show this pair from the top, side, front perspectives. The image is a cropped 3D volume (4096 nm x 4096 nm x 4096 nm around the center of the volume), so you should pay attention to discontinuities in the center of the image. Images are presented in groups of 3 (top, side, front). The segments merged together should look like a continuous single axon, where the orange segment is progressing in the same direction as the blue segment was progressing. They should just join together at the center; they shouldn't be overlapping.
If there is a split error and the proposed merge operation fixes it (the segments merged together look like a continuous single axon), then return 1. If there is no split error OR the merge operation is incorrect, then return -1.

Figure 3: Prompt used for identifying split error corrections. Text in blue is the additional context included in the "Description" prompts.

The second task we examined was the ability for LLMs to resolve split errors. Split errors occur when the segmentation algorithm inappropriately separates segments of the same neuron. While there are many different kinds of split errors, the most common we found were split errors in neuronal processes. Neuronal processes are the projections from the neuronal cell body that conduct signals through which neurons communicate; axons and dendrites are both neuronal processes.

To generate positive examples of split error corrections, we used the edit history of proofread segmentations where humans identified split errors in segment $s_i$ and found the correct segment $s_j$ to be merged. To generate negative examples of split error correction, we started by computing the "interface point" between $s_i, s_j$. We do this by computing the distances between the vertices of the $s_i, s_j$ meshes, identifying the shortest distances between points across the meshes, finding all points with a minimum distance threshold, and taking the average 3D coordinate. Then, we sampled a segment $s_k \neq s_j$ that would lead to incorrect merges by drawing from segments within 128 nanometers laterally and 120 to 880 nanometers vertically of the interface point. The range for the vertical dimension is due to missing imaging slices. To account for the fact that missing slices often lead to split errors due to the discontinuity of the imaging data, we have to adjust to find potential merge partners at the slice where the imaging is restored. We generate the images using a 4096 nm x 4096 nm x 4096 nm bounding box around the interface point to crop the images (see Figure 3 for example).

The distribution of split error correction examples is in Table 3. Our data generation procedure yields more positive than negative examples of split error corrections since the correct merge partner is occasionally the only segment within the available range. While all examples are available in the benchmark, we conduct our analysis on a random subset of 100 split error examples. For each example, we prompt the LLMs ten times and determine the final answer using majority vote. To provide an expert baseline, trained graduate or undergraduate students rated ≈50 examples for each condition. Additionally, we finetuned a ResNet-50 for an additional baseline (see Appendix G).

Table 3: # of Split and Merge Error Examples

|  | Split Error | | Merge Error | |
| --- | --- | --- | --- | --- |
|  | FlyWire | MICrONS | FlyWire | MICrONS |
| Positive Examples | 298 | 494 | 137 | 148 |
| Negative Examples | 248 | 473 | 137 | 148 |

### 4.2.1 Binary classification

With this data, our first version of the task was to prompt the LLM with images of a pair of segments and ask if the two segments should be merged to resolve a split error. We found for o4-mini and gpt-4o, the accuracy rate was above chance (=50%) but substantially lagged human performance (see Table 4). As is evident in the ROC curves in Figure 4, different models have different accept-reject biases. For instance, o4-mini rejects many potential merges, while Claude 4 Sonnet accepts nearly all of them. Adding information through the description has marginal effects on the performance in most cases, and a slightly negative effect on o4-mini.

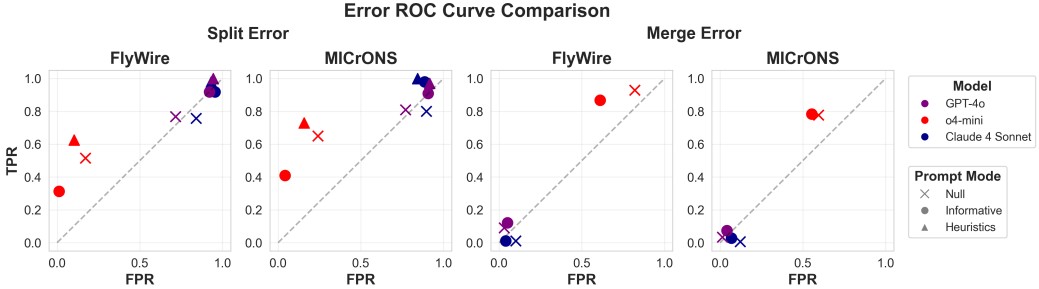

Figure 4: ROC Curves for the binary split error correction and merge error identification tasks. TPR=True Positive Rate, FPR=False Positive Rate.

### 4.2.2 Multiple choice classification

While the LLMs struggled with identifying correct versus incorrect split error corrections in a binary format, we were curious if they were better at multiple choice comparisons between different split error corrections. As such, we devised a second version of the task, where we show two candidate

merge partners for a segment and prompt the model to say which one (or neither) is a correct merge. In this case, as shown in Table 5, the proprietary models do much better than the null baseline, with o4-mini achieving 82.8% on FlyWire and 79.0% on MICrONS. Additionally, we find that adding the description information in the prompt significantly improves performance across both datasets for all models tested, except o4-mini which already showed strong performance.

### 4.2.3 Heuristics from LLM reasoning improves performance

Even for the best performing models, we wanted to understand how we could further improve on errors cases. As such, we analyzed the common error patterns found in the visual reasoning of o4-mini for the binary and multiple choice tasks on the MICrONS dataset. In our analysis, we found multiple assumptions in how o4-mini reasoned about split errors. For instance, the model assumed that proper merges needed to be small thin extension or that large gaps between segments necessarily meant the merges were not correct. However, both assumptions are often false; split segments are often the same size as their counterpart segment, and large gaps can come from artifacts like missing data. In response to these reasoning patterns, we developed seven "heuristics" that help guide the visual reasoning of the model to combat some of its own internal biases, and included these in the prompt in addition the base "Description" prompt. As a result, performance improved on binary classification and multiple choice across almost all models (see Tables 4 and 5, Figure 6). These findings demonstrate the potential of using the natural language reasoning ability of LLMs to both understand their failure cases and improve their performance.

Table 4: Performance on Split Error Correction Task (Binary)

| Model | FlyWire (95% CI) | MICrONS (95% CI) |
|---|---|---|
| Claude Sonnet 4+Null | 0.476 (0.406, 0.545) | 0.459 (0.388, 0.526) |
| o4-mini+Null | 0.663 (0.599, 0.733) | 0.704 (0.643, 0.765) |
| GPT-4o+Null | 0.540 (0.476, 0.610) | 0.526 (0.459, 0.597) |
| Claude Sonnet 4+Description | 0.508 (0.433, 0.578) | 0.556 (0.490, 0.628) |
| o4-mini+Description | 0.631 (0.567, 0.701) | 0.679 (0.612, 0.745) |
| GPT-4o+Description | 0.524 (0.449, 0.599) | 0.510 (0.444, 0.582) |
| Claude Sonnet 4+Heuristics | 0.551 (0.481, 0.626) | 0.587 (0.515, 0.658) |
| o4-mini+Heuristics | **0.754 (0.695, 0.813)** | **0.786 (0.724, 0.847)** |
| GPT-4o+Heuristics | 0.556 (0.487, 0.626) | 0.536 (0.469, 0.602) |
| Human | 0.840 (0.740, 0.940) | 0.902 (0.804, 0.980) |
| ResNet-50 ($\pm$ STD) | 0.720$\pm$0.034 | 0.667 $\pm$ 0.038 |

Table 5: Performance on Split Error Correction Task (Multiple Choice)

| Model | FlyWire (95% CI) | MICrONS (95% CI) |
|---|---|---|
| Claude Sonnet 4+Null | 0.475 (0.374, 0.576) | 0.530 (0.430, 0.620) |
| o4-mini+Null | 0.828 (0.747, 0.899) | 0.720 (0.630, 0.800) |
| GPT-4o+Null | 0.556 (0.465, 0.657) | 0.620 (0.520, 0.710) |
| Claude Sonnet 4+Description | 0.657 (0.566, 0.747) | 0.700 (0.610, 0.790) |
| o4-mini+Description | **0.828 (0.747, 0.909)** | 0.790 (0.710, 0.860) |
| GPT-4o+Description | 0.717 (0.626, 0.798) | 0.720 (0.630, 0.800) |
| Claude Sonnet 4+Heuristics | 0.677 (0.586, 0.768) | 0.770 (0.690, 0.850) |
| o4-mini+Heuristics | 0.788 (0.707, 0.869) | **0.850 (0.780, 0.910)** |
| GPT-4o+Heuristics | 0.667 (0.576, 0.758) | 0.720 (0.630, 0.800) |
| Human | 0.900 (0.820, 0.960) | 0.920 (0.840, 0.980) |
| ResNet-50 ($\pm$ STD) | 0.721$\pm$0.62 | 0.693$\pm$0.075 |

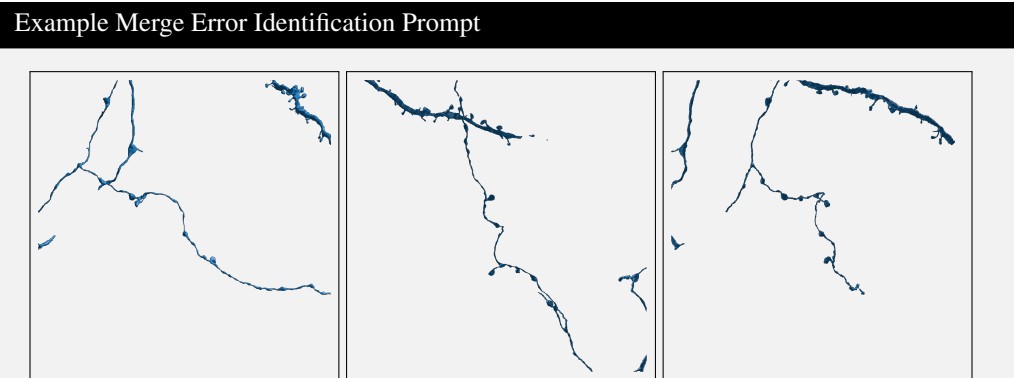

**Example Merge Error Identification Prompt**

You are an expert in analyzing neuronal morphology and split and merge errors in connectomics data. The previous images show a portion of 3D segmentation of neuronal data. While it's intended that the segment all correspond to processes (axon, dendrites) of a single neuron, it's possible that the algorithm may have introduced merge errors, inappropriately grouping processes from different neurons together. Merge errors are often characterized by aberrant axonal structure like the axon doubling back after branching off or an axon forming a ninety degree angle when joined with another. The images show this segment from the top, side, front perspectives. The image is a cropped 3D volume (8192 nm x 8192 nm x 8192 nm) around the center of the volume, so you should pay attention to merges in the center of the image. Images are presented in groups of 3 (top, side, front). If there is a merge error and the segment should be split apart, then return 1. If there is no merge error, then return -1.

Figure 5: Prompt used for identifying merge errors. Text in blue is the additional context included in the "Description" prompts.

## 4.3 Merge error identification

The third capability we examined was the ability for the LLMs to judge merge errors, which occur when the segmentation algorithms join segments from multiple different neurons. There are two scales of identifying merge errors. First, there are the merge errors identifiable from multiple somas appearing in the same segment. In Section 4.1, the precision and recall of the "multiple soma" category shows the performance of the LLMs at identifying multi-soma merge errors (see Appendix D for per-category precision and recall across models). Second, there are merge errors evident from some aberrant structure of the neuronal processes (i.e. axons or dendrites). Examples of aberrant structures include the axon doubling back after branching off or when one axon has an unnatural junction with another. Presently, we evaluate how well LLMs can identify the second kind of merge errors.

To generate examples of merge errors, we use the edit history of proofread segmentation to find segments where humans identified and corrected merge errors by introducing a split in the segmentation. We select for merge error corrections that resulted in two separate segments accessible through the CAVEClient interface. The coordinates where the split was introduced serves as the center point (x, y, z) of the generated 3D images. Then, we choose the smaller of the split segments, identify the size (width, height, depth) of the bounding box that encloses it, and set the new bounding box to be $((x - \text{margin}, y - \text{margin}, z - \text{margin}), (x + \text{margin}, y + \text{margin}, z + \text{margin}))$, where $\text{margin} = \max(4096 \text{ nm}, 2 * \max(\text{width}, \text{height}, \text{depth}))$. This variable bounding box strategy is implemented to provide an appropriate scale to reason about the neuronal processes. To generate negative samples, we use the same center point and bounding box but applied it to the final proofread 3D mesh. We assume that since they are proofread, all merge errors should be removed.

The distribution of merge error identification examples is in Table 3. While all examples are available in the benchmark, we conduct our analysis on a random subset of 100 examples, use majority voting, and provide expert and finetuned ResNet-50 baselines. As an additional baseline, we attempted to use another merge error detection method developed by Celii et al. [2025]. However, this method's requirements (i.e. the soma must be present in the segment) limited evaluation to only 14 examples

from our benchmark. We were not able to correctly identify errors in the 14 examples using this method (see Appendix H for further details).

Similar to split error corrections, we pursue binary and multiple choice versions of the task. For the binary version of the task, we prompt the LLMs to determine whether or not there is a merge error in the selected segment. The performance for most models is slightly above the null baseline (see Table 6). o4-mini stands out as the strongest model, achieving 62.8% and 61.5% accuracy on FlyWire and MICrONS, respectively, when also provided a description about merge errors.

Table 6: Performance on Merge Error Identification Task (Binary)

| Model | FlyWire (95% CI) | MICrONS (95% CI) |
|---|---|---|
| Claude Sonnet 4+Null | 0.457 (0.382, 0.533) | 0.443 (0.385, 0.500) |
| o4-mini+Null | 0.553 (0.477, 0.613) | 0.591 (0.534, 0.645) |
| GPT-4o+Null | 0.533 (0.462, 0.603) | 0.510 (0.453, 0.564) |
| Claude Sonnet 4+Description | 0.487 (0.412, 0.558) | 0.480 (0.426, 0.537) |
| o4-mini+Description | **0.628 (0.563, 0.693)** | **0.615 (0.557, 0.666)** |
| GPT-4o+Description | 0.538 (0.467, 0.608) | 0.517 (0.459, 0.571) |
| Human | 0.740 (0.620, 0.860) | 0.800 (0.680, 0.900) |
| ResNet-50 ($\pm$ STD) | 0.769$\pm$0.035 | 0.798 $\pm$0.02 |

For the multiple choice version of the task, we prompt the model to decide which one (or neither) of the two selected segments has a merge error. For both the FlyWire and MICrONS datasets, o4-mini stands out with the best performance (achieving 74.0% and 70.3% resp, see Table 7).

Table 7: Performance on Merge Error Identification Task (Multiple Choice)

| Model | FlyWire (95% CI) | MICrONS (95% CI) |
|---|---|---|
| Claude Sonnet 4+Null | 0.610 (0.545, 0.680) | 0.483 (0.426, 0.534) |
| o4-mini+Null | **0.740 (0.680, 0.800)** | 0.689 (0.635, 0.740) |
| GPT-4o+Null | 0.465 (0.400, 0.540) | 0.351 (0.301, 0.402) |
| Claude Sonnet 4+Description | 0.560 (0.490, 0.630) | 0.530 (0.476, 0.584) |
| o4-mini+Description | 0.670 (0.605, 0.730) | **0.703 (0.652, 0.750)** |
| GPT-4o+Description | 0.345 (0.285, 0.415) | 0.361 (0.311, 0.416) |
| Human | 0.840 (0.740, 0.940) | 0.796 (0.673, 0.898) |
| ResNet-50 ($\pm$ STD) | 0.569 $\pm$0.062 | 0.541 $\pm$0.018 |

## 5 Conclusion

In this paper, we introduced ConnectomeBench, a benchmark for evaluating LLMs' ability on three tasks important for connectome proofreading: segment identification, split error correction, and merge error identification. Our results show that current models can achieve surprisingly high performance in segment identification and both binary and multiple choice split error correction, though they struggle with merge error tasks. While these tasks do not capture all of the skills required for AI proofreading systems (e.g., synapse identification, merge error correction, etc.), they are critical skills for any such system. As LLMs continue to improve in their visual reasoning capabilities, we anticipate significant advances in their ability to assist and eventually replace human effort in connectome proofreading. ConnectomeBench provides a foundation for measuring progress toward this goal.

## Acknowledgments and Disclosure of Funding

ESB acknowledges HHMI, Lisa Yang, NIH R01AG087374, NIH 1R01EB024261, NIH 1R01AG070831, and John Doerr. JB acknowledges funding support from the Fannie and John Hertz Foundation.

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

## A Examples of Non-neuronal Type Segments

Example from FlyWire

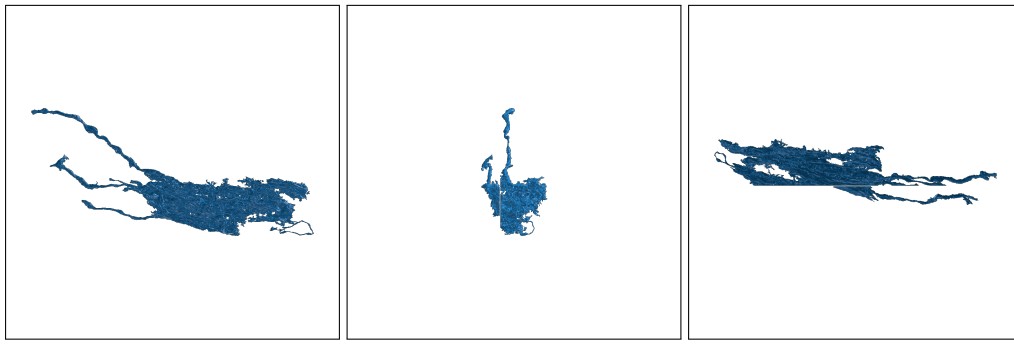

Example from MICrONS

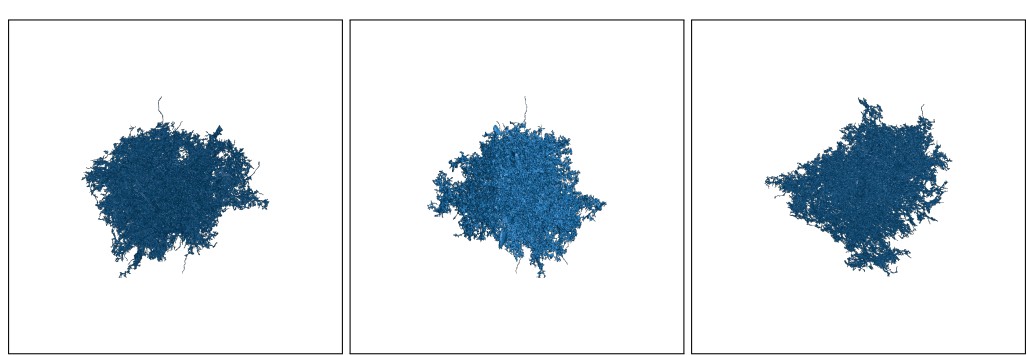

## B Split Error Correction Heuristics

- If the orange segment is taking up the complete (all you can see is orange) field of view and it's not spherical, the merge operation is not correct. Auto reject this option.
- If the orange segment is very small compared to the blue segment, the merge operation is not correct. Auto reject this option.
- If the orange segment is a sphere and the blue segment is not visible or is overlapping with the orange segment, the merge operation is correct.
- If the orange segment is a similar size to the blue segment at the interface point at the center of the image, then the merge operation is correct. Also, the orange segment can and often is a tube of similar volume: it doesn't need to be a small thin extension.
- If there is a big gap between the orange and blue segments at the center of the image, that's OK since it's likely that there are missing imaging planes. If the orange segment is going in the same direction as the blue segment was, it's an appropriate merge.
- If the orange and blue segments are parallel and lined up next to each other, then it's likely they are distinct processes of two different neurons. This is not a proper merge.
- Remember that you're reasoning in 3 dimensions. A segment might look short in one view, but long in another because of the perspective (looking at it dead on vs. from the side).

# C Accuracy comparison across prompt conditions

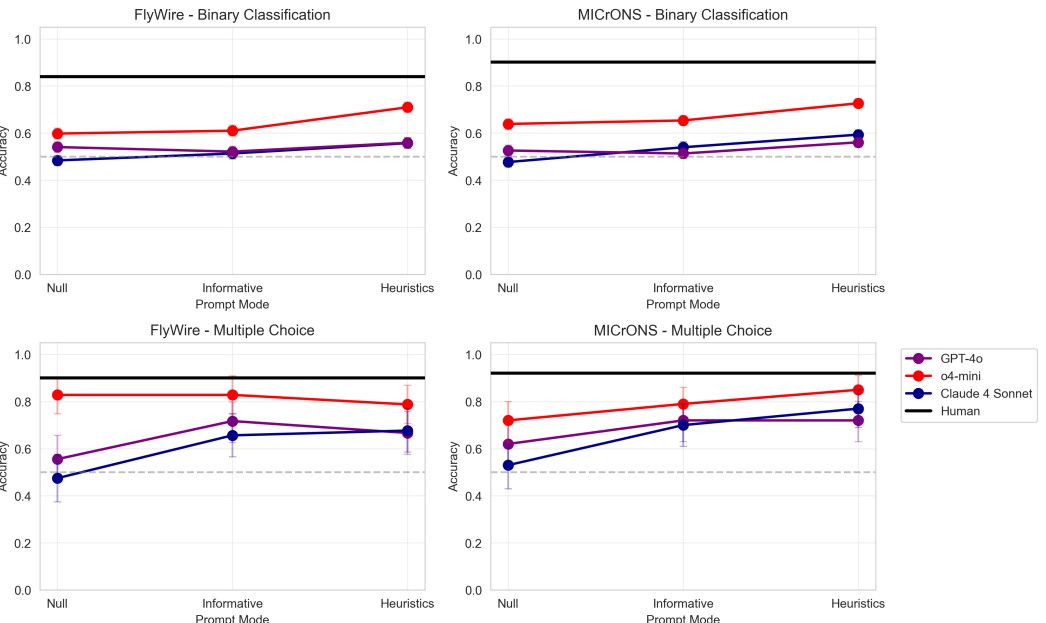

Figure 6

# D  Class specific accuracy, precision, and recall for segment identification

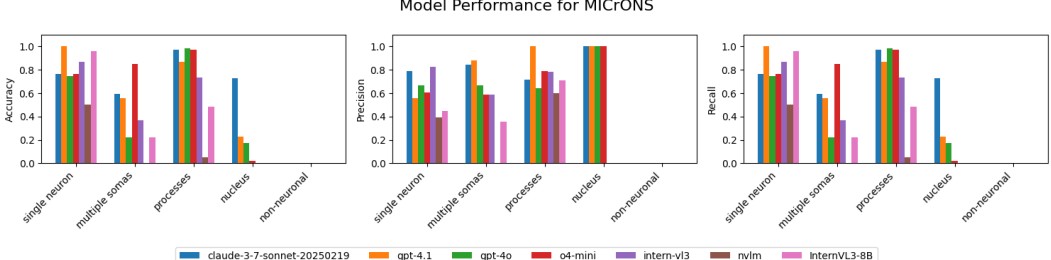

Figure 7: Accuracy, precision, and recall on segment classification. MICrONS, "Description"

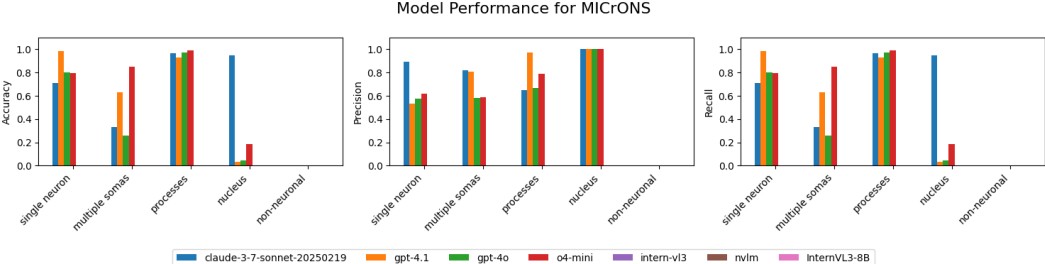

Figure 8: Accuracy, precision, and recall on segment classification. MICrONS, "Null"

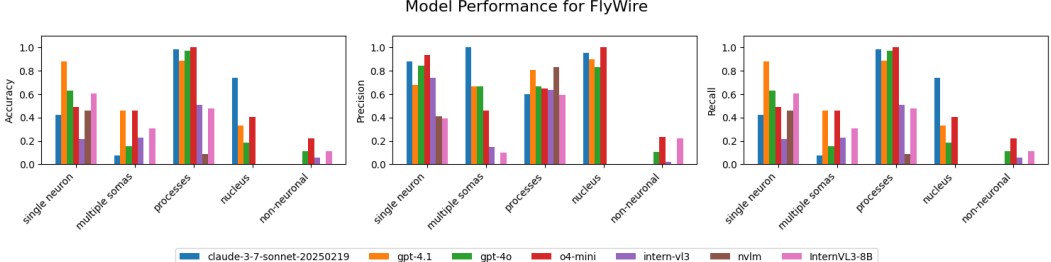

Figure 9: Accuracy, precision, and recall on segment classification. FlyWire, "Description"

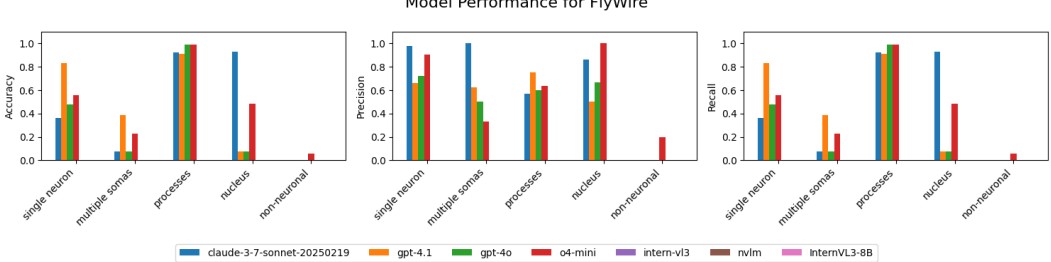

Figure 10: Accuracy, precision, and recall on segment classification. FlyWire, "Null"

# E Prompts

**Split Error Comparison Prompt (Description)**

1. Option ID {segment_id_1}
<image><image><image>
2. Option ID {segment_id_2}
<image><image><image>
You are an expert in analyzing neuronal morphology and merge errors in connectomics data. The previous images show a potential merge of a split error at the center of the 3D volume. Each option displays a pair of segments: the original segment (blue) and a potential merge candidate segment (orange). {image_description} below show this pair from the {view_description} perspectives for each option. The image is a cropped 3D volume ({2*zoom_margin} nm x {2*zoom_margin} nm x {2*zoom_margin} nm around the center of the volume), so you should pay attention to discontinuities in the center of the image. Images are presented in groups of {len(views)} ({view_description}) per option. The first group corresponds to Option 1, the second to Option 2, and so on. The segments merged together should look like a continuous single axon. They should just join together at the center; they shouldn't be overlapping. Pick the number (e.g., "1", "2", etc.) of the single best option that represents the correct merge. If none of the options show segments that should be merged, respond with "-1".
Surround your analysis with <analysis> and </analysis> tags. Surround your final answer (the number or "-1") with <answer> and </answer> tags.

**Split Error Comparison Prompt (Null)**

1. Option ID {segment_id_1}
<image><image><image>
2. Option ID {segment_id_2}
<image><image><image>
You are an expert in analyzing neuronal morphology and merge errors in connectomics data. The previous images show a potential merge of a split error at the center of the 3D volume. Each option displays a pair of segments: the original segment (blue) and a potential merge candidate segment (orange). {image_description} below show this pair from the {view_description} perspectives for each option. The image is a cropped 3D volume ({2*zoom_margin} nm x {2*zoom_margin} nm x {2*zoom_margin} nm around the center of the volume), so you should pay attention to discontinuities in the center of the image. Images are presented in groups of {len(views)} ({view_description}) per option. The first group corresponds to Option 1, the second to Option 2, and so on. Pick the number (e.g., "1", "2", etc.) of the single best option that represents the correct merge. If none of the options show segments that should be merged, respond with "-1".
Surround your analysis with <analysis> and </analysis> tags. Surround your final answer (the number or "-1") with <answer> and </answer> tags.

## F    Open Source Model Details

Our primary focus for open-weight models was on NVIDIA's NVLM and the InternVL-3 family, as a representative of leading open-weight multimodal architectures. Our experiments included two versions of the InternVL-3 family: the InternVL-3 8B (8 billion parameters) and the significantly larger InternVL-3 78B (78 billion parameters). For these open-source models, we achieved consistent output by setting `do_sample` to be false when configuring generation, thus negating the need for multiple runs per prompt by using greedy decoding. All computations for these models were performed on a system equipped with 4 NVIDIA H100 GPUs.

A key methodological aspect for the InternVL-3 models was their image preprocessing, specifically the "dynamic tiling" technique detailed in their documentation. This involves resizing an image and then dividing it into smaller patches (tiles) plus a thumbnail for model processing. We applied this standard tiling for the InternVL-3 8B model. However, due to substantial GPU memory constraints, this tiling step was omitted for the larger InternVL-3 78B model. Consequently, the 78B model processed images as single, un-tiled inputs, effectively operating at a lower input resolution than its standard configuration. This hardware-driven adaptation allowed the 8B model (with tiling) to serve as an indicator of tiling's general impact. Exploring whether full tiling could benefit the 78B model remains an area for future investigation.

To optimize throughput during these experiments, we leveraged the existing `batch_chat` functionality provided with the InternVL models. This feature enabled us to maximize batch sizes for both open-source models and process multiple instances concurrently. The computational time for each of the open-source model evaluations was approximately 2-4 hours.

## G    ResNet Training Details

We implemented ResNet-50 baselines for all connectomics tasks using ImageNet pretrained weights with task-specific input adaptations. The first convolutional layer was modified to handle varying input channels: 3 channels for segment classification (grayscale views stacked), merge comparison and identification (RGB images concatenated horizontally), and split identification (grayscale views stacked); 6 channels for split comparison (two neurons with 3 views each, grayscale stacked). All models used adaptive average pooling and replaced the final fully connected layer based on the number of classes per task. Training employed AdamW optimizer with learning rate 1e-4 for new classifier parameters and 1e-5 for pretrained backbone layers, weight decay 1e-4, and ReduceLROnPlateau scheduler (factor=0.5, patience=5 epochs). Input images were resized from 1024×1024 to 512×512 or 512×1536 and normalized using ImageNet statistics without data augmentation. Cross-entropy loss with balanced class weights addressed dataset imbalance, and weighted random sampling was used during training. Segment classification used an 80/20 train/validation split, while merge and split tasks employed 5-fold stratified cross-validation on an 80/20 train/test split with fixed random seed (42) for reproducibility.

## H    NEURD Baseline

This study reproduced the NEURD auto-proof split-suggestion workflow on the MICRONS dataset to evaluate its detection performance against a benchmark of pre-defined merge error events. The workflow operated on single neurons in non-interactive mode, processing nucleus-backed segments and mapping NEURD's geometric suggestions to event-level decisions based on spatial proximity.

### H.1    Infrastructure and Environment

The computational environment consisted of a Linux GPU virtual machine running Docker with NVIDIA runtime support. The NEURD software was deployed using the vendor-supplied container image celiib/neurd:v2 with GPU attachment enabled. The repository was mounted as the working directory, and NEURD was installed in editable mode within the container to mirror the tutorial configuration. A desktop-capable environment was maintained to support NEURD's container entrypoints and diagnostic tools.

Data access to the MICRONS public datastack (minnie65_public) was configured through the CAVE API. An access token was obtained after accepting the terms of service and provided to the container's CloudVolume integration. To reduce redundant mesh downloads, a host-side CloudVolume cache directory was mounted into the container, allowing mesh shards to be cached across runs.

## H.2   Cohort Selection and Event Definition

The analysis cohort comprised 14 unique nucleus-backed segments derived from benchmark event annotations. Nucleus-backed segments were selected to align with NEURD's design emphasis on soma-associated merge errors. Each event record in the benchmark specified a segment identifier, an interface point in nanometer coordinates representing the merge error location, and optional metadata including timestamps and operation identifiers.

## H.3   Workflow Implementation

NEURD's MICRONS data interface was initialized with default parameters for voxel scaling and API endpoints. The auto-proof stage (v7 filters) was invoked programmatically to perform mesh-driven decomposition, skeletonization, and rule-based filtering of potential split locations. The primary output collected was split_locations_before_filter, which contained coordinates of suggested split points prior to final filtering. Multi-soma suggestions were excluded from this analysis as they fell outside the nucleus-backed cohort definition.

An adapter module was developed to translate NEURD's per-neuron suggestions into per-event binary decisions compatible with benchmark metrics. For each segment, the adapter initialized the MICRONS interface, fetched the mesh representation, and attempted to load cached neuron objects from previous runs. When no cache was available, the mesh was optionally decimated, a neuron object was constructed, and auto-proof was executed. Suggestion coordinates were then extracted and deduplicated from the split_locations_before_filter output.

For each benchmark event, the adapter computed the minimum Euclidean distance in nanometers from the event's interface point to all suggestion coordinates for that segment. An event was classified as detected (neurd_detected equals true) if this minimum distance was at most 3000 nanometers. The adapter recorded the number of suggestions, minimum distance (or null if no suggestions existed), and execution time for each event.

## H.4   Caching Strategy and Performance Optimization

A two-tier caching strategy was implemented to reduce computational overhead. Pre-autoproof caches captured the neuron object state after preprocessing but before rule-based filtering, while post-autoproof caches captured the complete state including all outputs. The reuse policy prioritized post-autoproof caches when available, followed by pre-autoproof caches, with full reconstruction as a fallback. An option to require pre-autoproof cache presence and skip segments lacking it was provided for controlled reruns.

Mesh decimation was applied using pymeshlab to reduce face counts to approximately 1.5 to 2.0 million faces, decreasing preprocessing time and memory consumption while preserving geometric fidelity for rule evaluation. The final production run was performed without decimation to ensure maximum accuracy despite increased computational cost. An exposed parameter allowed auto-proof to run without the downstream after-statistics aggregation pass, which preserved split-location emission while avoiding aggregation failures observed in initial testing.

## H.5   Synapse Input Configuration

The workflow utilized live synapse access through NEURD's MICRONS data interface. For the nucleus-backed cohort, synapse materialization effectively returned empty results after filtering, causing runs to proceed in geometry-only mode with synapse counts and densities recorded as zero. The NEURD tutorial demonstrates an alternative approach using curated per-segment CSV synapse files to provide synaptic context, but this method was not employed in the present study.

## H.6 Execution and Monitoring

Execution was orchestrated by launching the container with NEURD installed and invoking the adapter with environment variables specifying event inputs, detection radius, MICRONS release name, cache directory, and decimation parameters. Boolean flags controlled nucleus filtering, cache reuse policy, and the after-statistics toggle. Segments were processed serially within a single container instance.

Monitoring infrastructure captured GPU telemetry (utilization percentage and memory usage) and container telemetry (CPU percentage, memory usage, and process counts) at periodic intervals, appending measurements to CSV files for time-series analysis. The adapter emitted structured logs documenting cache reuse decisions, decimation operations, stage start and finish markers, and suggestion counts. Per-segment timing summaries recorded initialization time, mesh fetch duration, neuron build time, auto-proof stage duration, and total elapsed time. Exceptions were logged with full message text for subsequent classification and debugging.

## H.7 Failure Handling and Stabilization

Initial executions with live synapse inputs encountered exceptions during or immediately after auto-proof execution, typically manifesting as incomplete feature frames or empty array concatenations following synapse filtering. These failures were addressed by disabling the final aggregation step (after-statistics), which retained the core rule-based filtering and split-location extraction while bypassing the aggregation operations that triggered exceptions. Following this modification, runs completed successfully and produced all expected outputs including adapter logs, timing summaries, and results in JSON format.

## H.8 Results and Resource Characterization

Pre-autoproof caches were successfully generated for all 14 segments in the cohort, with compressed file sizes typically in the tens of megabytes per segment. Post-autoproof caches were not produced in the described runs since cache saving was conditional on complete stage execution with all outputs.

Event-level results showed that all 14 nucleus-backed events yielded zero suggestions from NEURD, resulting in null values for minimum distance and false classification for all events at the 3000 nanometer detection radius. The summary statistics recorded 14 events across 14 segments with a hit rate of zero.

Resource utilization during preprocessing and auto-proof execution showed container CPU usage in the low triple-digit percentage range (indicating modest multi-core utilization), GPU utilization in the low single digits, and peak memory consumption in the low tens of gigabytes. Per-segment wall-clock execution times ranged from one to several hours depending on mesh complexity and cache availability.

## H.9 Reproducibility Parameters

The complete workflow can be reproduced using container image celiib/neurd:v2 against the MICRONS public datastack with the recorded release name. The detection radius was fixed at 3000 nanometers, and all coordinates were maintained in nanometer units throughout the pipeline with optional voxel-to-nanometer scaling applied to mesh representations. The caching policy prioritized pre-autoproof cache reuse with an option to require pre-cache presence. Mesh decimation was configurable via pymeshlab with face targets specified in run parameters.

