# OpenReview forum: "ConnectomeBench: Can LLMs proofread the connectome?"
_NeurIPS.cc/2025/Datasets_and_Benchmarks_Track — NeurIPS 2025 Datasets and Benchmarks Track spotlight_

### Official Review · Reviewer_Un6m · 2025-06-25

**Rating:** 4
**Confidence:** 1

**Summary:**

This paper presents ConnectomeBench, a novel benchmark for evaluating LLMs on connectome proofreading tasks.  It provides a
standardized method for evaluating LLM capabilities in connectome proofreading. The representation is clear and the motivation is well-explained. Several LLMs are evaluated in the experiments.

**Dataset Code Accessibility:**

Yes

**Dataset Code Comments:**

The benchmark is available

**Ethical Considerations:**

No, there are no or only very minor ethics concerns

**Limitations Weaknesses:**

Actually, I am not an expert in this field. Some of my concerns may be due to my lack of understanding of this field. My main concerns are as follows:
1. The benchmark focuses only on 3/10+ known proofreading error types. Critical tasks like synapse identification are omitted, potentially overstating practical applicability.
2. In Table 2, why does Claude Sonnet 3.7 obtain inconsistent performances in FLyWire and MICrONS? Besides, in Table 6, o4-mini leads in merge error detection without a clear explanation.
3. The computational constraints preventing full dynamic tiling for InternVL-3 78B may underrepresent open-source capabilities. The >50% accuracy thresholds claimed as "surprisingly high" lack empirical justification compared to human performance.

**Strengths Contributions:**

1. The introduction of standardized tasks (segment identification, split/merge error correction) fills an important gap in evaluating LLMs for connectomics. The use of real-world datasets (MICrONS and FlyWire) provides ecological validity.
2. The systematic comparison of proprietary (Claude 3.7, GPT-4 variants) and open-source models (InternVL) across multiple prompting strategies offers meaningful insights. The ROC curves in Figure 3 effectively visualize model performance tradeoffs.
3. Detailed methodology for data generation and the careful analysis of per-category performance (Appendix A figures) demonstrate thorough experimental design.

---

> ### Author Rebuttal · Authors · 2025-07-31
>
> Thank you for reviewing our paper!
>
> Limited scope of proofreading error types
>
> As you note, we focus on only a subset of the important tasks in proofreading and omit critical tasks like synapse identification. We acknowledge these limitations in our paper. Our goal was specifically to profile LLMs' 3D mesh reasoning capabilities to assess the potential of LLM-based proofreading agents. While there are established ML models for synapse identification in both mouse and drosophila brain, we could not find a benchmark for these merge/split errors to test LLMs. Moreover, we're excited by the prospect of building LLM proofreading agents that can use tools like established synapse identification models in future work.
>
> Inconsistent performance across datasets
>
> The meshes between FlyWire and MICrONS datasets have different characteristics, and our best hypothesis is that MICrONS meshes better align with Claude 3.7's visual priors for the shapes of neurons.
>
> o4-mini specific performance on Merge Error identification
>
> We looked into the reasoning traces of o4-mini and found that it understands the visual scenes better than other models. When we analyzed 100 examples from split error correction tasks, o4-mini could identify spatial relationships between segments and handle 3D perspective effects more reliably than other models. More tellingly, o4-mini was the only model that actually benefited from targeted heuristics we developed from error analysis. These heuristics improved o4-mini's accuracy from 65.0% to 87.0% on mouse data and from 56.0% to 79.0% on fly data. The same heuristics did nothing for Claude 4 Sonnet, suggesting o4-mini has better foundational scene understanding which might contribute to superior performance on the merge error identification task. In our revisions, we will include our analysis of the visual reasoning fidelity of o4-mini vs other models.
>
> Computational constraints for open-source models
>
> Your concern about computational constraints potentially underrepresenting open-source capabilities, particularly for InternVL-3 78B where we couldn't implement full dynamic tiling, is valid and represents important future work. We acknowledge this limitation and believe that properly optimized implementations of large open-source models could potentially achieve better performance than our current results suggest.
>
> Justification for "surprisingly high" accuracy claims
>
> We have now conducted comprehensive human baseline experiments for each category, showing that human experts achieve 70.0-96.1% accuracy across different tasks.
>
> | Task                               | Mouse - Human Annotation   | Fly - Human Annotation     |
> |------------------------------------|---------------------------|---------------------------|
> | Split Error Correction (Pointwise)| 90.1% (CI: 78.9% - 95.7%) | 84.0% (CI: 71.6% - 91.6%) |
> | Split Error Correction (Pairwise) | 95.9% (CI: 86.5% - 98.8%) | 96.1% (CI: 86.8% - 98.9%) |
> | Merge Error Detection (Pointwise) | 76.4% (CI: 63.2% - 85.9%) | 70.0% (CI: 56.4% - 80.8%) |
> | Merge Error Detection (Pairwise)  | 84.0% (CI: 71.5% - 91.7%) | 72.7% (CI: 59.2% - 83.0%) |
>
> With these benchmarks established, LLM performance of 60-87% represents a more substantial fraction of human-level performance than we initially expected.

---

> > ### Comment · Area_Chair_2oxo · 2025-08-04
> > **Reviewer response due**
> >
> > Dear reviewer,
> >
> > There are only a couple of days left in the discussion period, and your engagement during this time is critical. Please provide your response to the author rebuttal as soon as possible, this is key for having a meaningful reviewer-author discussion and providing the authors the opportunity to discuss and provide clarification on any of your remaining concerns.
> >
> > Thank you!

---

### Official Review · Reviewer_9gYa · 2025-07-02

**Rating:** 4
**Confidence:** 4

**Summary:**

This paper benchmarks a lot of closed and open models on the proofreading three types of connectome problems: identification, split error, and merge error on two well studied connectome datasets: MICrONS and FlyWire The results show that identification performances are generally higher than split error and merge error tasks. This indicates that LLMs still have a big improvement space for such a human proofreading tasks in the scientific field of neural connectomics.

**Dataset Code Accessibility:**

Yes

**Ethical Considerations:**

No, there are no or only very minor ethics concerns

**Final Justification:**

I don't have a lot of questions regarding this submission. Generally speaking, I think it is a good paper to be accepted by the NeurIPS dataset/benchmark track. The topic is interesting to me, and I think it is worth being previewed by a lot of computational neuroscience people. So, I'll keep my score 4 as borderline acceptance.

**Limitations Weaknesses:**

* As mentioned in the prior work section, there are graph-based models. I think at least there should be one traditional algorithm, some deep neural network models like CNN or GNN as baselines when comparing with the LLMs. Can authors also show one result from e.g., Celii et al. [2025]? Besides, since this is a connectome related tasks, are there any graph neural network (GNN) based baseline models?
* The ground truth proofreading results are mainly based on previous works. Therefore, this might limit the understanding of how well the benchmark will scale to new datasets, especially when there are not only 2 types of errors mentioned in the current work.
* Question: Line 135, how authors determine "the most common answer", manually or by some algorithm? A little bit more details here would be better.
* In all tables, it would be better to bold the best model.

**Strengths Contributions:**

* The paper is well presented, with clear experiments setup, work flow charts, analysis procedures, and clear results. The three main tasks: segment identification, split error, and merge error are common challenging tasks for softwares and human post-hoc validations. Therefore, this work provides an important insight of how LLM can perform on such tasks in replacement of existing traditional algorithms, and how reliable it can reduce the human's workload.
* The experiments are comprehensive, with different ways of prompting the LLMs and evaluations and across several popular LLMs in recent days.
* The contribution of creating and dealing such datasets for benchmarking is meaningful and interesting. Specifically, this is the first benchmark that systematically evaluates large language models (LLMs) on connectome-based reasoning tasks across by multi-modality inputs, bridging neuroscience and AI.

---

> ### Author Rebuttal · Authors · 2025-07-31
>
> Thank you for reviewing our paper!
> In short, we prompt the language model to return an answer multiple times, count the frequency of each answer, and choose the most frequent answer as the response for that prompt. All of this is done in our post processing script. Also, yes, we will bold the best models!

---

> > ### Comment · Reviewer_9gYa · 2025-08-04
> >
> > I thank the author for the rebuttal. I have two further concerns.
> >
> > First, ss mentioned in the prior work section, there are graph-based models. I think at least there should be one traditional algorithm, some deep neural network models like CNN or GNN as baselines when comparing with the LLMs. Can authors also show one result from e.g., Celii et al. [2025]? Besides, since this is a connectome related tasks, are there any graph neural network (GNN) based baseline models?
> >
> > Second, the ground truth proofreading results are mainly based on previous works. Therefore, this might limit the understanding of how well the benchmark will scale to new datasets, especially when there are not only 2 types of errors mentioned in the current work.

---

> > > ### Author Response · Authors · 2025-08-06
> > >
> > > Thank you for the follow-up. We address your concerns below.
> > >
> > > Regarding baselines: Since the original submission, we have added ResNet and Human baselines across all tasks. We shared these results with other reviewers but hadn't included them in our response to you since they weren't among your original concerns. The results are summarized below.
> > >
> > > | Task                               | Mouse ResNet      | Fly ResNet        | Mouse Human           | Fly Human             |
> > > |------------------------------------|-------------------|-------------------|-----------------------|-----------------------|
> > > | Split Error Correction (Pointwise)| 79.79% ± 2.04%    | 76.86% ± 3.59%    | 90.1% (78.9% - 95.7%) | 84.0% (71.6% - 91.6%) |
> > > | Split Error Correction (Pairwise) | 54.11% ± 1.84%    | 56.86% ± 6.20%    | 95.9% (86.5% - 98.8%) | 96.1% (86.8% - 98.9%) |
> > > | Merge Error Detection (Pointwise) | 66.67% ± 3.80%    | 72.00% ± 3.37%    | 76.4% (63.2% - 85.9%) | 70.0% (56.4% - 80.8%) |
> > > | Merge Error Detection (Pairwise)  | 69.33% ± 7.42%    | 72.14% ± 6.14%    | 84.0% (71.5% - 91.7%) | 72.7% (59.2% - 83.0%) |
> > >
> > > We will include results from Celii et al. [2025] in the camera-ready version. Note that their method is designed for merge error detection and does not address split errors, so comparisons will be limited to the pointwise merge error detection tasks.
> > >
> > > Regarding GNN baselines: While there are emerging GNN-based approaches for connectome proofreading (such as the Allen Institute's deep-neurographs), we could not find published performance evaluations/models that would enable fair comparison. Additionally, our benchmark focuses on visual reasoning over 3D neuron segment images, which requires identifying topological errors from visual cues in the segmented volumes. This is conceptually different from graph-based approaches that operate on connectivity patterns between neuron fragments. We acknowledge that exploring GNN approaches could be valuable future work as these methods mature.
> > >
> > > Regarding generalization concerns: We agree that relying on existing ground truth annotations poses scalability questions. However, we designed our benchmark using two distinct species (mouse and fly) with different neuron morphologies and error patterns. The fact that the strongest models generalize across both datasets despite these structural differences suggests they are reasoning in a generalizable way rather than simply memorizing dataset-specific patterns. This cross-species generalization is also part of the reason we used interpretable heuristics rather than fine-tuning, as we wanted to test the models' inherent visual reasoning capabilities over this kind of data. While we currently focus on split error correction and merge error identification (two of the most common and impactful error types in connectome proofreading), we hope that this visual reasoning abilities extends to other task classes (like merge error correction, 3d navigation/manipulation across neuron mesh(es) during proofreading, etc). Moreover, we are expanding the benchmark with new public datasets while collaborating with connectomics labs to incorporate additional fully-annotated datasets as they become available.

---

> > > > ### Comment · Reviewer_9gYa · 2025-08-07
> > > >
> > > > Thanks for these explanations. I don't have further questions and I think this is a innovative work to be published after including Celii et al. [2025]. I'll keep my supporting score.

---

### Official Review · Reviewer_nYvS · 2025-07-02

**Rating:** 5
**Confidence:** 3

**Summary:**

The paper introduces and describes ConnectomeBench, a benchmark designed to evaluate the model's ability to proofread the connectome. The benchmark has three tasks: segment type identification, split error correction and merge error identification, which correspond to the three tasks that human researchers normally have to for this task. Authors formulate the problem in a zero-shot setting, giving the model a prompt plus a collection of images of the 3d meshes, and measure the accuracy for each task. The paper evaluates several proprietary LLMs (GPT 4o, 4.1, o4-mini and Claude Sonnet 3.7) and open-access LLMs (InternVL, NVLM).

**Additional Feedback:**

Minor concern: inconsistent use of \citet vs \citep, e.g. L49 "Dorkenwald et al. [2024]", L89 "Tavakoli et al. [2025]" possibly missing parentheses or need paraphrasing. Many similar typos throughout the paper.

**Dataset Code Accessibility:**

Yes

**Dataset Code Comments:**

The code is generally in good shape. The instructions are clear, the codebase itself appears reasonably well-structured.

Minor question: do you need *all* these libraries in requirements.txt or did you just dump your entire environment there? If the latter is the case, consider cleaning it up.

The dataset card in https://huggingface.co/datasets/jeffbbrown2/ConnectomeBench appears to be rather minimalistic (just a link to the repo). While it is up to the authors do choose what to do with it, it would be nice to at least consider some basic directions there.

**Ethical Comments:**

## Possible mismatch in the use of human subjects (aka "undergrad students are made of people!")

L164 says "Afterwards, **trained undergraduates and graduate students went through each example** to classify and describe the 3d mesh"

L622 checklist "Crowdsourcing and research with human subjects" says "There was **no crowdsourcing** and research with human subjects done in this work."

To the best of my understanding, **grad students are (technically) humans**, and so there appears to be a contradiction between the two statements. If I misunderstood the nature of their role and this does not count as crowdsourcing, I request that authors clarify it in their response.

**Note to AC:** I did not flag this as an ethics consideration since I believe that this is an honest mistake of either the paper authors or myself. If you disagree, please either tell me to flag it or flag it yourself, whichever you prefer.

**Ethical Considerations:**

No, there are no or only very minor ethics concerns

**Final Justification:**

I believe that the additional error analysis and justifications provided in the author response go a long way to justify the setup and evaluation metric. I also appreciate simplifying the prompt, though it was not the deciding factor.

**Limitations Weaknesses:**

While the paper overall presents a good benchmark, there are several questionable setup choices that would, in my opinion, necessitate ablation analysis.

## On the input format

In L122, you describe that you opted to present the problem using the images of 3d meshes.

While I am not a domain expert, it appears that there are alternative ways to represent the input, such as:
- an interactive view that allows the LLM to choose and shift their viewing angle as a form of tool use
- text graph representations (e.g. vertices and edges) and possibly others.

My concern is that choosing the suboptimal representation can limit the model's ability to solve the problem. There are two ways to ablate this: either try all reasonable representations and check if there is a difference, or see if the representation you use is sufficient for humans.

In the latter case, you can prompt trained humans with the same prompt as you do LLMs *instead of giving them their usual tools* and see if they can solve the problem equally well to their usual score. If not, consider changing the input format.

## Prompt design

Your prompts (e.g. Fig. 1 and 2) include certain 'affirmations' that resemble how people prompt non-instruct models (e.g. "You are an expert in analyzing ..."). Do these help the performance of the LLMs in question? If not, consider removing them. Similarly, it would be best to ablate any other design choices in hopes of simplifying the benchmark.

## Error metric and context

Authors often report LLM scores in terms of accuracy, often in the 60-80% range. What are the practical implications of having this accuracy? For instance, how do they compare against human accuracy scores in the same setup?

If the current models' accuracy does not allow for practical application in place of humans, what of other use types? For instance, a model that is not accurate enough to replace a human can still be used to aid a human moderator, e.g. by creating a short list of cases that require human attention, thereby speeding up the annotation process.

If the latter application type is more realistic, consider switching to more appropriate quality metrics, e.g. recall / precision @ k or whatever makes practical sense.

## Error analysis

At least some of the models used in this study are capable of reasoning, which would normally help understand the cause of their errors. Based on the reasoning traces (and any other information authors find appropriate), **what kind of mistakes do the LLM annotators tend to make?** Do they struggle in understanding the image, lack professional knowledge, or fail to make logical implications? (or, perhaps, another reason entirely) Is this distribution of mistakes similar or different to how human annotators fail? Do LLMs consistently make the same mistake on repeated API calls or do they err in different ways? (if the latter is the case, consider ensembling / voting)

**Strengths Contributions:**

The paper introduces an interesting (and, to my knowledge, at least somewhat new) problem for LLM evaluation that can have immediate practical applications. The approach to constructing the benchmark appears generally sound (barring several concerns in the next section). The paper is generally well-written, well-structured and easy to follow. The supplementary code has clear instructions and includes the exact dependency versions.

---

> ### Author Rebuttal · Authors · 2025-07-31
>
> Thank you for reviewing our paper! We hope to address your concerns below.
>
> On the input format
>
> We intentionally focused on static 3D mesh images to appraise LLMs' performance relative to humans in this specific visual reasoning domain. Our goal was to test the models' ability to understand these 3D mesh representations as a necessary step before considering more complex setups like interactive views that would allow the LLMs to, for instance, select/deselect meshes, shift viewing angles as a form of tool use, or utilize EM data directly.
>
> To establish this comparison, we conducted human baseline experiments where we worked with the same static mesh images provided to the LLMs and rated 50-51 examples per task:
>
> | Task                               | Mouse Human           | Fly Human             |
> |------------------------------------|----------------------|-----------------------|
> | Split Error Correction (Pointwise)| 90.1% (CI: 78.9% - 95.7%) | 84.0% (CI: 71.6% - 91.6%) |
> | Split Error Correction (Pairwise) | 95.9% (CI: 86.5% - 98.8%) | 96.1% (CI: 86.8% - 98.9%) |
> | Merge Error Detection (Pointwise) | 76.4% (CI: 63.2% - 85.9%) | 70.0% (CI: 56.4% - 80.8%) |
> | Merge Error Detection (Pairwise)  | 84.0% (CI: 71.5% - 91.7%) | 72.7% (CI: 59.2% - 83.0%) |
>
> This establishes a clear watermark for when the models are capable enough to visually reason in the restricted case of only 3D meshes. The goal in follow-up work is to arm LLM agents with the full toolset for proofreading, but this work was seen as a necessary step to determine when and which models could be ready for use in that context.
>
> Prompt design
>
> We conducted the exact ablation the reviewer suggested. Removing the "You are an expert in analyzing..." preamble did not change performance at all, so we will remove it in the final version.
>
> Error metric and context
>
> The reviewer's question about practical implications is important. Our honest assessment is that we want to evaluate how well the models perform on these tasks relative to humans to determine which models are worthwhile to train into specialized agents for proofreading. The 60-80% accuracy range we currently observe doesn't enable direct practical application, but it does suggest that some models have the visual reasoning priors necessary to make specialized training worthwhile. This benchmark serves as a screening tool to identify promising base models before investing in fine-tuning for connectomics applications.
>
> Error analysis
>
> Thank you for this comment! In response, we conducted error analysis on the mistakes made in 100 examples from the split error correction task using o4-mini. By examining the model's reasoning traces, we identified common failure patterns and developed eight specific heuristics:
>
> 1. If the orange segment is taking up the complete (all you can see is orange) field of view and it's not spherical, the merge operation is not correct. Auto reject this option.
> 2. If the orange segment is very small compared to the blue segment, the merge operation is not correct. Auto reject this option.
> 3. If the orange segment is a sphere and the blue segment is not visible or is overlapping with the orange segment, the merge operation is correct.
> 4. If the orange segment is a similar size to the blue segment at the interface point at the center of the image, then the merge operation is correct. Also, the orange segment can and often is a tube of similar volume: it doesn't need to be a small thin extension.
> 5. If there is a big gap between the orange and blue segments at the center of the image, that's OK since it's likely that there are missing imaging planes. If the orange segment is going in the same direction as the blue segment was, it's an appropriate merge.
> 6. If the orange and blue segments are parallel and lined up next to each other, then it's likely they are distinct processes of two different neurons. This is not a proper merge.
> 7. Remember that you're reasoning in 3 dimensions. A segment might look short in one view, but long in another because of the perspective (looking at it dead on vs. from the side).
> 8. If the orange and blue segments are overlapping globular shells, then auto accept this merge operation.
>
> Implementing these heuristics in the prompt improved o4-mini's accuracy from 65.0% to 87.0% on mouse data and from 56.0% to 79.0% on fly data when tested on the remaining data not used for heuristic development. So, for this model, the issue seems to be the model did not have enough "professional knowledge" to accomplish the task. However, these same heuristics did not improve Claude 4 Sonnet's performance. When examining Claude 4 Sonnet's reasoning traces, it was clear the model was not understanding the images as well as o4-mini, suggesting that the effectiveness of such heuristics depends critically on the underlying model's visual understanding capabilities.
>
> In this particular case, we find that sometimes o4-mini makes the same mistakes across API calls and sometimes it does not. Interestingly, we found that response consistency strongly correlates with accuracy on the mouse split error correction using o4-mini with heuristics:
>
> Low confidence (≤0.6): Accuracy 0.733, F1 0.778
> Medium confidence (0.6-0.8): Accuracy 0.829, F1 0.829
> High confidence (1.0): Accuracy 0.955, F1 0.947
>
> Concerning the ethics comment: All of the people that classified the meshes were authors on the paper, so to the best of my knowledge this doesn't count as crowdsourcing. If that's not the case, happy to change it!

---

> > ### Comment · Area_Chair_2oxo · 2025-08-04
> > **Reviewer response due**
> >
> > Dear reviewer,
> >
> > There are only a couple of days left in the discussion period, and your engagement during this time is critical. Please provide your response to the author rebuttal as soon as possible, this is key for having a meaningful reviewer-author discussion and providing the authors the opportunity to discuss and provide clarification on any of your remaining concerns.
> >
> > Thank you!

---

### Official Review · Reviewer_rrY2 · 2025-07-03

**Rating:** 5
**Confidence:** 3

**Summary:**

The paper introduces ConnectomeBench, a benchmark for connectomics proofreading using off-the-shelf multimodal large language models (LLMs). Proprietary LLMs including Claude 3.7 Sonnet, o4-mini, GPT-4.1, GPT-4o, and open-weight LLMs including InternVL-3 and NVLM were used to perform three proofreading tasks (segment type identification, split error correction, and merge error detection) on the FlyWire and MICrONS datasets. The results show some promises as well as pitfalls of using LLMs to complement human experts in connectomics proofreading tasks.

**Dataset Code Accessibility:**

Yes

**Dataset Code Comments:**

The dataset was uploaded to HuggingFace and the code was publicly available on Github. They seemed to be well documented and didn't raise any concerns for me.

**Ethical Considerations:**

No, there are no or only very minor ethics concerns

**Final Justification:**

The paper made meaningful contributions to the research on proofreading in connectomics by providing a comprehensive benchmark where LLMs are leveraged for the proofreading task. I initially had some concerns on the lack of non-LLM baselines, lack of qualitative analyses and missing confidence intervals on the results; however, the authors have addressed them nicely in the rebuttal. I think the paper would be much improved with the inclusion of these results in the final version of the paper. I'm happy to raise my score and recommend acceptance.

**Limitations Weaknesses:**

1. While the paper focuses entirely on LLM baselines, it would benefit greatly from including more non-LLM baselines such as CNN-based approaches mentioned in the Prior Work section. Without these baselines, it is difficult to assess whether LLMs significantly outperform conventional methods to justify their computational cost.
2. The results lack confidence intervals which are essential to gauge the significance of the performance across models.
3. Although properly acknowledged as a limitation in Section 5 of the paper, I believe comparing off-the-shelf LLMs with fine-tuned LLMs would add more values to the paper. Even a small study where the most performant LLM is finetuned on a reasonable ground truth set would be good to gauge the promises of the finetuned LLMs.
4. Some qualitative analyses showing examples where LLMs succeed or fail would be interesting to take a look at and see if there’s any consistent patterns there, which could in turn inform us to construct better prompts that guide the LLMs to avoid failure modes.
5. Minor: texts on Figure 3 are pretty small and difficult to read.

**Strengths Contributions:**

1. The paper proposes the use of LLMs in connectomics proofreading, which helps advance the curation of large-scale neuroscientific datasets and has great implications for the field if proven to be effective.
2. The choice of the datasets (MICrONS and FlyWire) to evaluate the LLM capabilities is well justified by the scale of these datasets.
3. The writing is generally clear, well structured, and easy to follow.

---

> ### Author Rebuttal · Authors · 2025-07-31
>
> Thank you for reviewing our paper!
>
> 1. Lack of non-LLM baselines
>
> Your concern about the lack of baselines is well taken. To address the concerns, we implemented CNN-based baselines using ResNet-50 architecture with 5-fold cross-validation, as well as human expert baselines. The results are summarized below:
>
> | Task                               | Mouse - ResNet      | Fly - ResNet        | Mouse - Human Annotation           | Fly - Human Annotation             |
> |------------------------------------|-------------------|-------------------|-----------------------|-----------------------|
> | Split Error Correction (Pointwise)| 79.79% ± 2.04%    | 76.86% ± 3.59%    | 90.1% (78.9% - 95.7%) | 84.0% (71.6% - 91.6%) |
> | Split Error Correction (Pairwise) | 54.11% ± 1.84%    | 56.86% ± 6.20%    | 95.9% (86.5% - 98.8%) | 96.1% (86.8% - 98.9%) |
> | Merge Error Detection (Pointwise) | 66.67% ± 3.80%    | 72.00% ± 3.37%    | 76.4% (63.2% - 85.9%) | 70.0% (56.4% - 80.8%) |
> | Merge Error Detection (Pairwise)  | 69.33% ± 7.42%    | 72.14% ± 6.14%    | 84.0% (71.5% - 91.7%) | 72.7% (59.2% - 83.0%) |
>
> Human baselines were evaluated on 50-51 random examples per task category. For ResNet implementation, we used 3 channels for RGB values in the split error correction tasks. For pairwise split error correction we concatenated the images horizontally. For merge error correction, we used grayscale with channels accommodating both the 3 views and the number of options. The substantially worse performance on pairwise split tasks may be attributable to the horizontal concatenation approach, which likely disrupts the spatial relationships that CNNs rely on for feature extraction.
>
> 2. Missing confidence intervals
>
> In response to the concerns over the confidence intervals, we have now computed 95% confidence intervals for all our results using bootstrap sampling. We will include these confidence intervals in all results tables in the revised manuscript to properly assess the significance of performance differences across models.
>
> 3. No fine-tuned LLM comparison
>
> We agree this would strengthen the paper. While we have not yet completed fine-tuning experiments, our prompt optimization study demonstrates the potential for improvement even without doing so- we achieved accuracy improvements of 22% for mouse and 23% for fly datasets on the pointwise split error correction problem through systematic prompt engineering alone (see the next section). While the best models and prompts are leagues above our initial baseline, they still make fundamental mistakes in their 3-dimensional reasoning, suggesting that fine-tuned models could achieve even better performance by learning the specific spatial relationships in connectomics data. We are conducting fine-tuning experiments and will include results if space permits in the revision.
>
> 4. Missing qualitative analysis
>
> This is an excellent suggestion. We conducted a qualitative analysis on 100 examples from the split error correction task to identify common failure patterns. This analysis revealed eight specific failure modes and led us to develop corresponding heuristics to prompt the model.
>
> 1. If the orange segment is taking up the complete (all you can see is orange) field of view and it's not spherical, the merge operation is not correct. Auto reject this option.
> 2. If the orange segment is very small compared to the blue segment, the merge operation is not correct. Auto reject this option.
> 3. If the orange segment is a sphere and the blue segment is not visible or is overlapping with the orange segment, the merge operation is correct.
> 4. If the orange segment is a similar size to the blue segment at the interface point at the center of the image, then the merge operation is correct. Also, the orange segment can and often is a tube of similar volume: it doesn't need to be a small thin extension.
> 5. If there is a big gap between the orange and blue segments at the center of the image, that's OK since it's likely that there are missing imaging planes. If the orange segment is going in the same direction as the blue segment was, it's an appropriate merge.
> 6. If the orange and blue segments are parallel and lined up next to each other, then it's likely they are distinct processes of two different neurons. This is not a proper merge.
> 7. Remember that you're reasoning in 3 dimensions. A segment might look short in one view, but long in another because of the perspective (looking at it dead on vs. from the side).
> 8. If the orange and blue segments are overlapping globular shells, then auto accept this merge operation.
>
> Implementing these heuristics on o4-mini improved accuracy from 65.0% (without any heuristics) to 87.0% on mouse data and from 56.0% (without any heuristics) to 79.0% on fly data, with F1-scores improving from 0.478 to 0.866 (mouse) and 0.353 to 0.784 (fly), when tested on examples not used for prompt optimization. Notably, these improvements put the optimized LLMs ahead of the fine-tuned ResNet baselines. Interestingly, we found these heuristics did not immediately improve Claude 4 Sonnet (which we ran to include in our revisions), and examination of the reasoning traces revealed that Claude 4 Sonnet had inferior image understanding compared to o4-mini. This suggests that models that reliably perceive the data can be directly benefited from introducing interpretable heuristics. We will include this qualitative analysis in the revision as it provides valuable insights into model-specific capabilities and practical guidance for prompt optimization.
>
> 5. Figure 3 readability
>
> We will increase the font size and improve the clarity of Figure 3 in the revised manuscript.

---

> > ### Comment · Area_Chair_2oxo · 2025-08-04
> > **Reviewer response due**
> >
> > Dear reviewer,
> >
> > There are only a couple of days left in the discussion period, and your engagement during this time is critical. Please provide your response to the author rebuttal as soon as possible, this is key for having a meaningful reviewer-author discussion and providing the authors the opportunity to discuss and provide clarification on any of your remaining concerns.
> >
> > Thank you!

---

> > ### Comment · Reviewer_rrY2 · 2025-08-07
> >
> > Thank you for the responses to my concerns. I'm glad to see the improvements after targeting the common failure patterns, as well as the addition of comparison to CNN-based baselines. Regarding the missing confidence interval, could you show some results with confidence intervals that you have compiled so far? I understand that running these experiments takes time, so even initial results in one proofreading task would be fine for me to understand the performance variance of these LLMs.

---

> > > ### Author Response · Authors · 2025-08-07
> > >
> > > Yes, thank you for being understanding. Here, we use majority voting/consensus with 10 LLM samples for each prompt, and the results below are evaluated on a subset of 100 of the pointwise split error correction prompts. We're showing Claude 4 Sonnet instead of 3.7 since changes to the rate limit are making it even slower (though we can show both in the camera ready version). To get the confidence intervals, we use 1000 sample bootstrapping to get 95% confidence intervals. We also include pass@1 performance for comparison. Let us know if you would like to see anything else!
> > >
> > > MICrONS, cons@10
> > > | Condition   | Claude 4 Sonnet       | GPT 4.1               | GPT 4o                | o4-mini               |
> > > |-------------|----------------------|----------------------|----------------------|----------------------|
> > > | Null        | 48.0% (39.0% - 57.0%) | 50.0% (40.0% - 59.0%) | 43.0% (33.0% - 52.0%) | 65.0% (56.0% - 74.0%) |
> > > | +Heuristics | 56.0% (47.0% - 64.0%) | 53.0% (43.0% - 62.0%) | 51.0% (41.0% - 60.0%) | 87.0% (80.0% - 93.0%) |
> > >
> > > MICrONS, pass@1
> > > | Condition   | Claude 4 Sonnet       | GPT 4.1               | GPT 4o                | o4-mini               |
> > > |-------------|----------------------|----------------------|----------------------|----------------------|
> > > | Null        | 57.0% (49.2% - 64.4%) | 52.0% (44.8% - 59.0%) | 47.0% (39.3% - 54.6%) | 62.0% (55.0% - 69.7%) |
> > > | +Heuristics | 59.0% (52.3% - 66.2%) | 52.0% (44.4% - 59.1%) | 57.0% (49.3% - 64.1%) | 71.0% (64.2% - 77.8%) |

---

> > ### Comment · Reviewer_rrY2 · 2025-08-08
> >
> > Thank you for providing the results. This has resolved my question. It would be great if you can repeat this on other tasks and include them in the final revision of the paper. I'm happy to increase my score.

---

### Decision · Program_Chairs · 2025-09-18

**Decision:**

Accept (spotlight)

**Comment:**

**Strengths**

- The paper provides a novel benchmark to systematically evaluate LLMs' performance on three proofreading tasks (segment type identification, split error correction, and merge error detection) on the FlyWire and MICrONS datasets
- The paper was praised for being well-written, clearly structured, and easy to follow. The experimental setup, workflow and the accompanying dataset (and associated description) were also appreciated for their clarity by reviewers rrY2, nYvS, 9gYa.
- The systematic comparison across several popular proprietary and open-source LLMs using different prompting strategies was considered thorough and valuable (9gYa, Un6m).

In the rebuttal period several key additions were made to the paper making it more comprehensive (experiments with baseline non-LLM models such as CNNs), rigorous (ablation experiments to understand impact of instruction prompt on model outcomes) and insightful (qualitative error analysis to understand failure modes of current LLMs). Thank you to both the reviewers and authors for their productive contributions to the paper.

I encourage the authors to add in all the points discussed in the rebuttal to the updated manuscript and thereby address the questions raised and the clarifications sought by the reviewers.

**Weaknesses**
The human baselines and the qualitative error analysis added in the rebuttal period are quite crucial to the completeness of the paper. It would be great if the qualitative analysis could be further expanded and the insights generated from it should be discussed in relation to (1) existing literature on these tasks and the errors in baseline automated methods, (2) errors made by human experts in this space. Both these points would add rich observations and concretely aid future research in this space.

Provide the most important reasons for your decision to accept/reject.
-- Novel benchmark in a field, promoting innovation and new research
-- Rigorous experiment design and comprehensive results with further deliberation (qual error analysis) on room for improvement by LLMs in this task

===== FINAL UPDATE FROM DB Track PCs ====

The final decision for this paper has been taken by the program chairs after consultation with the SACs. All Senior Area Chairs have ranked papers according to the feedback from the AC during the review process. We decided to leave the original meta-review to reflect the opinion of the AC in light of the initial discussions with reviewers and SAC.